elifesciences.org eLIFE

# FGF mediated MAPK and PI3K/Akt Signals make distinct contributions to pluripotency and the establishment of Neural Crest

Lauren Geary[1], Carole LaBonne[1,2]*

[1]Department of Molecular Biosciences, Northwestern University, Evanston, United States; [2]Robert H Lurie Comprehensive Cancer Center, Northwestern University, Evanston, United States

**Abstract** Early vertebrate embryos possess cells with the potential to generate all embryonic cell types. While this pluripotency is progressively lost as cells become lineage restricted, Neural Crest cells retain broad developmental potential. Here, we provide novel insights into signals essential for both pluripotency and neural crest formation in *Xenopus*. We show that FGF signaling controls a subset of genes expressed by pluripotent blastula cells, and find a striking switch in the signaling cascades activated by FGF signaling as cells lose pluripotency and commence lineage restriction. Pluripotent cells display and require Map Kinase signaling, whereas PI3 Kinase/Akt signals increase as developmental potential is restricted, and are required for transit to certain lineage restricted states. Importantly, retaining a high Map Kinase/low Akt signaling profile is essential for establishing Neural Crest stem cells. These findings shed important light on the signal-mediated control of pluripotency and the molecular mechanisms governing genesis of Neural Crest.

DOI: https://doi.org/10.7554/eLife.33845.001

*For correspondence:
clabonne@northwestern.edu

Competing interests: The authors declare that no competing interests exist.

## Introduction

The evolutionary transition from simple chordate body plans to complex vertebrate body plans was driven by the acquisition of the Neural Crest, a unique stem cell population with broad, multi-germ layer developmental potential (*Le Douarin and Kalcheim, 1999*; *Hall, 2000*; *Bronner and LeDouarin, 2012*; *Prasad et al., 2012*). At gastrula and neurula stages, Neural Crest cells are found within the presumptive ectoderm at the neural plate border, and they will ultimately contribute to ectodermal derivatives, including much of the peripheral nervous system. However, despite their site of origin, neural crest cells also contribute many mesodermal cell types to the body plan, including cartilage, bone, and smooth muscle, and also make contributions to otherwise endodermal organs such as the thyroid (*Le Douarin and Kalcheim, 1999*). Until recently, models for how neural crest cells acquire their remarkably broad potential proposed that inductive interactions orchestrated by BMP, FGF, and Wnt signals endowed these cells with greater potency than the cells they were derived from developmentally or evolutionarily (*Huang and Saint-Jeannet, 2004*; *Taylor and LaBonne, 2007*; *Prasad et al., 2012*; *Rogers et al., 2012*; *Stuhlmiller and García-Castro, 2012a*). Such a mechanism conflicts, however, with the generalized view of embryonic development as a progressive restriction of developmental potential.

We recently demonstrated that much of the regulatory network which controls the pluripotency of blastula inner cell mass/animal cap cells is shared with neural crest cells, shedding new light on the origins of the neural crest cells and the evolution of vertebrates (*Buitrago-Delgado et al.,*

*2015*). Indeed, we found that factors that have long been considered neural crest potency factors, such as Snail1 (*Taylor and LaBonne, 2007*) and Sox5 (*Nordin and LaBonne, 2014*), are expressed earlier, in blastula animal pole cells, and are required for their pluripotency. Together, these findings suggest that neural crest cells arise through retention of the transcriptional circuitry that controls the pluripotency of the blastula cells they are derived from, avoiding the lineage restriction that characterizes neighboring cells (*Buitrago-Delgado et al., 2015*; *Hoppler and Wheeler, 2015*; *Le Douarin and Dupin, 2016*). This revised model raises fundamental questions about how the cells that will become the neural crest escape lineage restriction in order to maintain broad developmental potential, and how this relates to signals that have previously been implicated in the genesis of these stem cells. For example, BMP signaling has been found to play an essential role in the pluripotency of both blastula stem cells and neural crest cells (*Ying et al., 2003*; *Kléber et al., 2005*; *Nordin and LaBonne, 2014*), together with Sox5, which directs the target specificity of BMP R-Smads in both cell types (*Nordin and LaBonne, 2014*). It will be important to build on these insights and further delineate the roles of other signaling pathways in the retention of pluripotency.

FGF signaling is used reiteratively throughout embryonic development to pattern multiple tissue types and germ layers. While FGF signaling has a well established role in the formation of mesoderm, it has also been linked to the formation of the neuroectoderm/neural plate, as well as to anterio-posterior patterning of the CNS (*Slack et al., 1987*; *Amaya et al., 1991*; *Xu et al., 1997*; *Hongo et al., 1999*; *Hardcastle et al., 2000*; *Ribisi et al., 2000*; *Fletcher et al., 2006*; *Dorey and Amaya, 2010*; *Wills et al., 2010*). While anterior neural induction mediated by BMP antagonists can occur independent of FGF signaling, FGFs clearly play a role in posterior neural development (*Wills et al., 2010*). Importantly, FGF signaling has also been implicated in the establishment of both neural crest stem cells and the neural plate border region more generally (*Mayor et al., 1997*; *LaBonne and Bronner-Fraser, 1998*; *Monsoro-Burq et al., 2003*; *Hong et al., 2008*; *Stuhlmiller and García-Castro, 2012b*; *Yardley and García-Castro, 2012*). Although the precise role of FGF signaling in the establishment of these cell populations relative to BMP or Wnt signals is not fully understood, at least some enhancers for neural plate border genes have been shown to require FGF signaling (*Garnett et al., 2012*). Intriguingly, these same genes are also expressed in pluripotent blastula cells, making it important to re-examine the role of FGF-mediated signals at earlier times in *Xenopus* development, with a focus on understanding their role in the retention of blastula stage pluripotency proposed to underlie genesis of Neural Crest cells. Such studies might also help shed light on the highly context dependent role played by FGF signaling in the regulation of pluripotency in cultured embryonic stem (ES) cells (*Lanner and Rossant, 2010*). Activation of FGF signaling in naïve mouse embryonic stem cells (mESCs) promotes lineage restriction of these cells (*Kunath et al., 2007*), whereas FGF activity maintains primed embryonic stem cells, also known as epiblast stem cells (EpiSCs) in a pluripotent/undifferentiated state (*Brons et al., 2007*; *Tesar et al., 2007*). Given our model that neural crest cells arise via retention of the attributes of pluripotent blastula cells, we wondered if FGF signaling might play a role in preventing premature lineage restriction of these cells.

In this study, we investigate the requirement for FGF signaling in the transient pluripotency of blastula animal pole cells, and the subsequent establishment of the neural crest state. We find that FGF signaling is essential for normal gene expression in pluripotent blastula cells, and for the capacity of these cells to respond properly to lineage restriction cues. We investigate which FGF-dependent signaling cascades mediate these effects, and find a striking switch in cascade utilization as cells transit from a pluripotent state to a lineage restricted state. We show that pluripotent blastula cells exhibit high Map Kinase signaling, whereas cells undergoing lineage restriction are characterized by increased PI3 Kinase/Akt signaling. Finally, we provide evidence that the balance of FGF-directed Map Kinase and PI3 Kinase/Akt signaling activity plays a role in the retention of blastula stage potential in neural crest cells.

## Results

### FGF signaling is required for proper gene expression in pluripotent blastula cells

Because FGF signaling is known to play a role in the establishment of the neural crest cell population at the neural plate border in *Xenopus,* and is also linked to the control of pluripotency in mESCs, we sought to determine if these signals were required in the pluripotent animal pole cells of blastula stage embryos. Consistent with such a role, FGF receptor 4 (FGFR4) is expressed throughout the animal hemisphere of blastula stage embryos, where the pluripotent stem cells reside. By gastrula stages (St. 12), *FGFR4* expression is heightened in the neural plate border region, and by neurula stages (St. 15) is strongly enriched in neural crest forming regions of the embryo (*Figure 1a*). The expression pattern of FGFR4 at gastrula and neurula stages has been previously described (*Hongo et al., 1999*; *Golub et al., 2000*; *Lea et al., 2009*), and its expression in neural crest forming regions at neurula stages has been reported to overlap with that of Snail2 (*Golub et al., 2000*) in agreement with our unpublished observations.

To determine whether *FGFR4* expression correlates with the stem cell state, we utilized explants of pluripotent blastula stem cells ('animal caps'). At blastula stages, these explants are pluripotent and can be induced to give rise to any embryonic cell type. The pluripotency of these cells is transient in culture, however, as it is in the developing embryo. As explants age from blastula to gastrula then neurula stages, they lose pluripotency and become lineage restricted; in the absence of exogenous signals, they will transit to an epidermal state. We therefore examined expression of *FGFR4* in these explants as they aged. We found that at blastula stages, when explanted cells are pluripotent, they strongly express *FGFR4* (*Figure 1b*), however, as these cells transit to an epidermal state, *FGFR4* expression is lost. This expression pattern is consistent with a role for this receptor in events prior to the onset of lineage restriction. However, although *FGFR4* is the most abundant FGFR in these cells (unpublished data), other FGFRs are also expressed (*Lea et al., 2009*) and thus could play roles in control of pluripotency and lineage restriction.

In order to determine whether FGF signaling plays a role in pluripotency and lineage restriction in *Xenopus*, we used a dominant-negative inhibitory receptor to carry out loss of function studies. We chose a dominant negative strategy because we were interested in the overall role of FGF-mediated signals, rather than role of any specific receptor. While we mainly deployed a dominant negative FGFR4 for these studies, dominant negative receptors frequently inhibit the activity of other related receptors and its effects should therefore be interpreted as effects on FGF signaling in general, not on FGFR4 signaling specifically. Embryos expressing a dominant-negative FGFR4 (dnFGFR4) were cultured to blastula stages and examined by in situ hybridization for genes expressed by pluripotent animal pole cells. We found that blocking FGF signaling led to a significant reduction in the expression of both *Vent2* (98%, n = 186) and *Id3* (96%, n = 84), but did not alter expression of other factors such as *Myc* or *FoxD3* (*Figure 1c*). The requirement for FGF signaling for proper gene expression in pluripotent animal pole cells suggests an essential role for these signals in controlling the developmental state of these cells. We similarly found that cells expressing dnFGFR4 were deficient in their ability to give rise to neural crest cells in whole embryos, as assayed by expression of *FoxD3*, *Sox9* and *Snail2* (*Figure 1—figure supplement 1*). By contrast, we found that expression of a dnFGFR1 did not similarly lead to loss of neural crest formation, suggesting that these two dominant negative receptors have distinct activities. We thus utilized dnFGFR4 to block FGF signaling for subsequent experiments, while recognizing that it may block FGF receptors other than FGFR4.

### Blocking FGF signaling in pluripotent blastula cells interferes with the adoption of an epidermal state

Since blocking FGF signaling inhibited expression of *Vent2* and *Id3* in pluripotent blastula cells, we hypothesized that FGF signaling might be required for the pluripotency of these cells, and/or for cells to exit pluripotency and transit to a restricted state. At blastula stages, when cells are pluripotent, animal pole explants express core pluripotency markers such as *Sox2/3*, and the Oct4 homologue, *Oct60* (*pou5F3.3*). These genes are subsequently down-regulated as explants age and become restricted to an epidermal state. To test the requirement for FGF signaling in this process, embryos injected with dnFGFR4 were allowed to develop until blastula stages, when animal cap

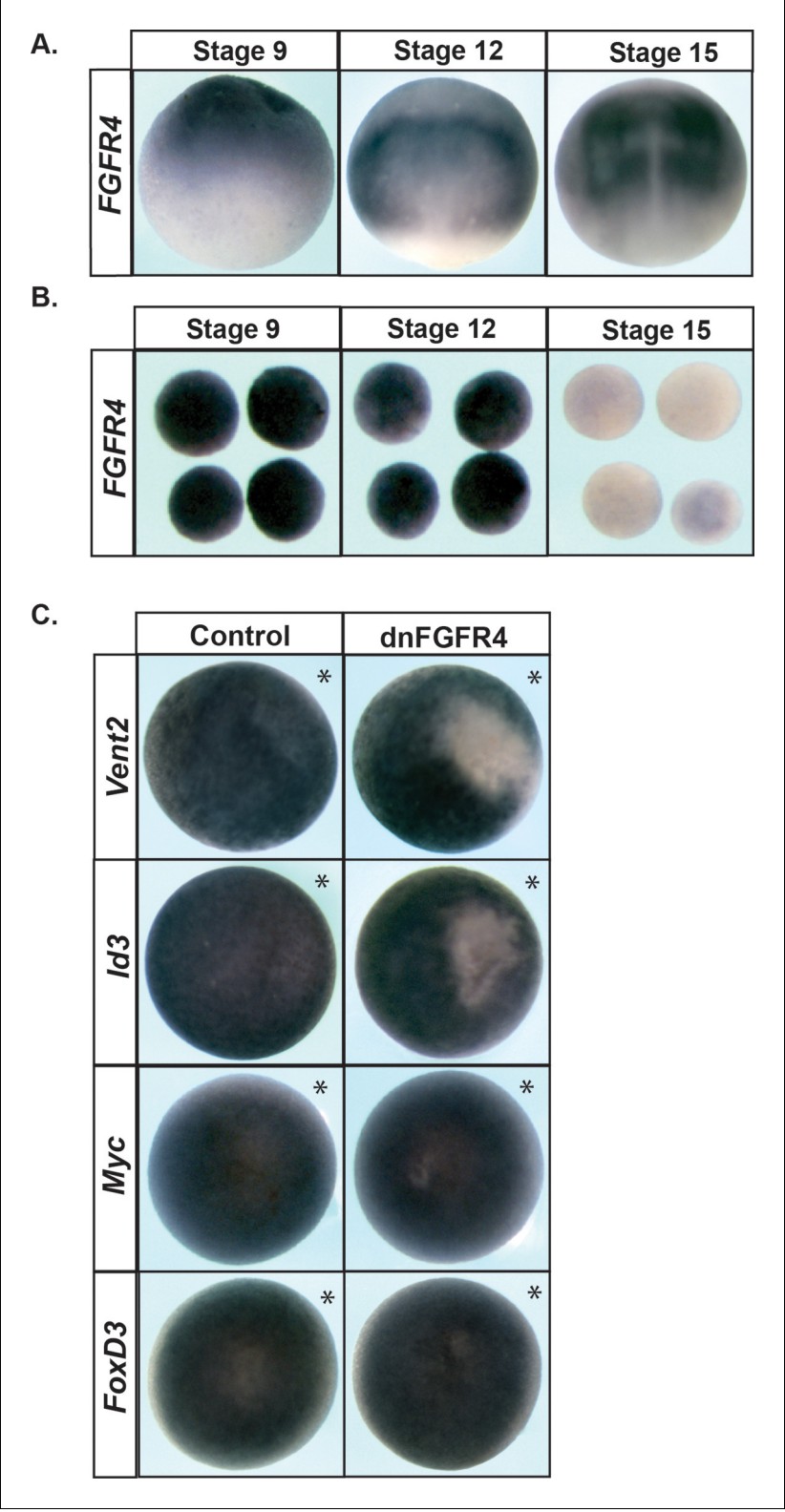

**Figure 1.** FGF signaling is required for proper blastula stage gene expression. (**A**) In situ hybridization examining *FGFR4* expression in wildtype *Xenopus* embryos collected at blastula (stage 9, lateral view, animal pole up), late gastrula (stage 12, dorsal view, anterior up), and mid-neurula (stage 15, dorsal view, anterior up) stages. Expression is seen in the pluripotent cells of the animal hemisphere at blastula stages and in the neural plate and neural crest forming regions at gastrula and neurula states. (**B**) Animal pole explant assay examining *FGFR4* expression.
*Figure 1 continued on next page*

*Figure 1 continued*

Explants were cultured alongside sibling embryos and collected at blastula (stage 9), late gastrula (stage 12), and mid-neurula (stage 15) stages. (C) In situ hybridization examining *Vent2, Id3, Myc,* and *FoxD3* expression in blastula stage (stage 9) embryos injected with dominant-negative FGFR4 (dnFGFR4). Asterisk denotes injected side, marked by staining of the lineage tracer β-galactosidase (red). Dominant-negative FGFR4 blocks expression of *Vent2* and *Id3*.

DOI: https://doi.org/10.7554/eLife.33845.002

The following figure supplement is available for figure 1:

**Figure supplement 1.** Blocking FGF signaling using dnFGFR4, but not dnFGFR1, leads to a loss in neural crest gene expression at mid-neurula stages.

DOI: https://doi.org/10.7554/eLife.33845.003

explants were isolated, and then cultured until sibling embryos reached blastula (St.9), gastrula (St.11) or neural plate (St.13) stages. We found that explants blocked for FGF signaling exhibited prolonged, low level expression of *Sox3* (95%, n = 97), and poorly expressed the epidermal marker *Epidermal Keratin (EPK)* (90%, n = 62), suggesting that FGF signaling was essential for pluripotent blastula cells to transit to an epidermal state (*Figure 2a,c*). Since *Sox3* expression is characteristic of both pluripotent cells and neuronal progenitor cells, we investigated whether explants expressing dnFGFR4 were being retained in a pluripotent state or instead were being biased toward a neuronal progenitor fate. We found that Stage 13 explants blocked for FGF signaling do not express the pluripotency factor *Oct60* (*pou5F3.3*), suggesting that they are not retaining pluripotency (*Figure 2b*).

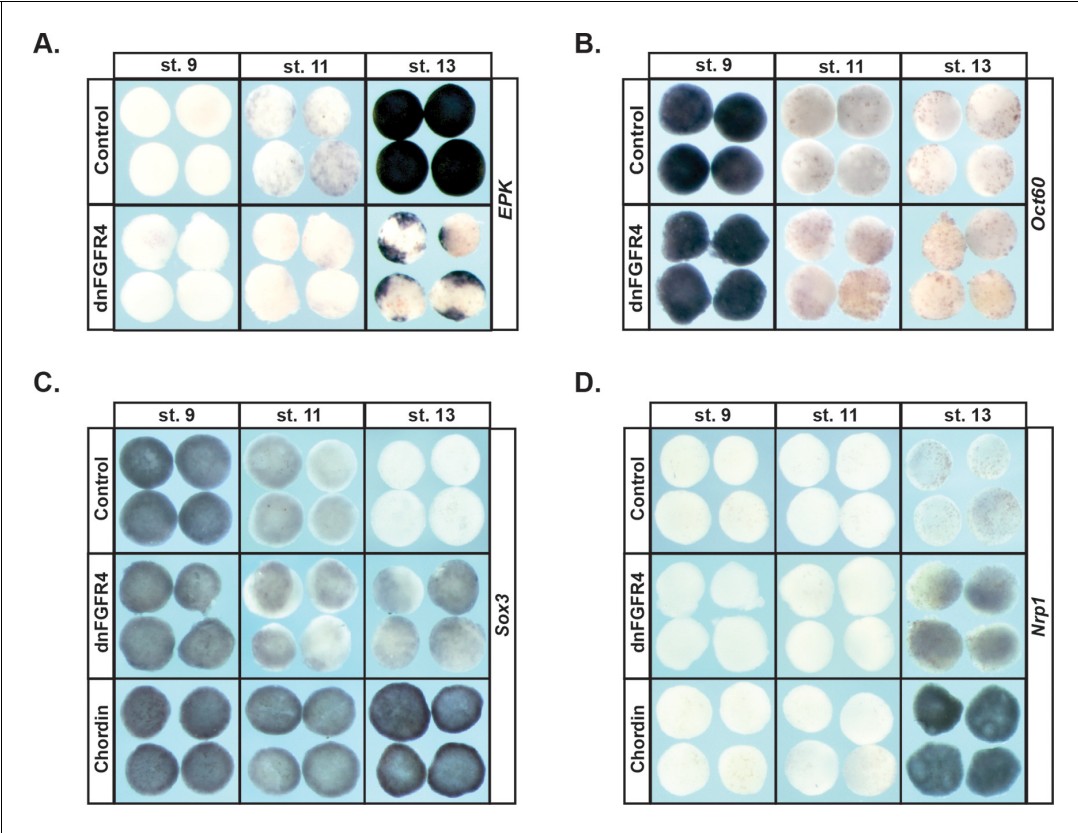

**Figure 2.** Blocking FGF signaling in pluripotent blastula cells interferes with adoption of an epidermal state and neuralizes cells. (A–D) In situ hybridization examining expression of *Epidermal Keratin (EPK)* (A) *Oct60* (B) *Sox3* (C) or *Nrp1* (D) in animal pole explants injected with dnFGFR4 or chordin for phenotypic comparison. Explants were cultured alongside sibling embryos and collected at blastula (stage 9), midgastrula (stage 11), and early neurula (stage 13) stages. Blocking FGF signaling interferes with *EPK* expression and mildly induces *Nrp1* expression.

DOI: https://doi.org/10.7554/eLife.33845.004

These explants do weakly express the definitive neural marker *Nrp1* (94%, n = 85), suggesting that they may be biased toward a neuronal progenitor state. However, the explants do not express levels of either *Sox3* or *Nrp1* associated with chordin-mediated neural induction, indicating that they are not adopting a definitive neural state (*Figure 2c,d*). It is possible that, as previously reported, FGF signals are later involved in the commitment of these cells to a neuronal state in response to neural-inducing cues (*Hongo et al., 1999*). Our experiments do not address this question.

## Cells progress from a high pERK state to a high pAkt state as they transit from the pluripotent to the lineage-restricted state

Like other tyrosine kinase receptors, FGF receptors can activate multiple downstream signaling cascades upon ligand binding, including the Ras/Map Kinase cascade that leads to Erk phosphorylation and activation, and the PI3 Kinase cascade that leads to Akt phosphorylation and activation (*Figure 3a*). Both of these signaling cascades have previously been implicated in Neural Crest development (*Stuhlmiller and García-Castro, 2012b*; *Pegoraro et al., 2015*). Given our findings that FGF signaling plays an essential role in pluripotent blastula cells, we wished to determine which signaling cascades were activated by FGF signaling in these cells during the transition from pluripotency to lineage restriction.

To assess the activation of these two cascades, we utilized antibodies that detect the phosphorylated, active, forms of Map Kinase and Akt. Animal pole explants were isolated at blastula stages and collected at blastula, gastrula, or early neural plate stages for western analysis. Unexpectedly, we detected a striking transition in cascade activation as cells transited from a pluripotent to a lineage-restricted state. Pluripotent cells of blastula-stage explants exhibited robust phosphorylated-Erk (pErk), which was lost as cells became lineage-restricted (*Figure 3b*). By contrast, pluripotent cells showed low or undetectable phosphorylated-Akt (pAkt), but activation of this kinase increased as cells lost pluripotency and transited to an epidermal state. Both the early pErk signal and the later pAkt signal were blocked in explants expressing dnFGFR4, confirming that both were FGF signaling-dependent (*Figure 3c*).

Because the above findings implicate two different signaling cascades in FGF-mediated regulation of pluripotency and lineage restriction, and because these signaling cascades appear to play temporally distinct roles in this transition, we wished to determine the contributions that each pathway makes to the developmental potential of these cells. To address this, we utilized reagents that can block the activation of each pathway. Map Kinase signaling was blocked using a chemical inhibitor of the upstream kinase Mek, RDEA119 (*Iverson et al., 2009*), ('Meki') or using a dominant negative form of Raf1 ('dnRaf'). Activation of Akt signaling was blocked by over-expressing a dominant-negative PI3 Kinase subunit (Δp85, 'dnPI3K') (*Carballada et al., 2001*; *Nie and Chang, 2007*) or using a chemical inhibitor of PI3 kinase, LY294 or Wortmannin ('PI3Ki'). We confirmed that these inhibitors blocked the activation of its respective cascade without interfering with the other pathway (*Figure 3—figure supplement 1a,b*). We then compared the effects of each of these inhibitors to dnFGFR4 in animal pole explants. We found that blocking Map Kinase signaling phenocopied the effects of blocking FGF signaling, preventing cells from transiting to an epidermal state, as evidenced by a loss of two different epidermal markers *EPK* (85%, n = 56) and *Trim29* (100%, n = 25) (*Figure 3d*). By contrast, blocking PI3 Kinase/Akt signaling with dnPI3K or with the chemical inhibitor LY294 had no effect on *EPK* or *Trim29* expression (*Figure 3d* and not shown). These findings reveal a differential requirement for Map Kinase and PI3 Kinase/Akt signaling during the transition from a pluripotent to an epidermal state.

## PI3K/AKT signaling, but not MAPK signaling, is required for pluripotent blastula cells to adopt a neural fate

Given the strikingly different responses of pluripotent animal pole cells to blocking Map Kinase vs. PI3 Kinase/Akt signaling with respect to adopting an epidermal state, we wished to examine the role these pathways play in adopting an alternative ectoderm-derived state, neuronal progenitor cells. It is well established that blocking BMP signaling with BMP antagonists such as Chordin directs cells to form neural plate rather than epidermis (*Sasai et al., 1995*; *Zimmerman et al., 1996*). We therefore examined the effects of blocking Map Kinase or PI3 Kinase/Akt on Chordin-mediated neural induction. Chordin expressing animal pole explants, but not control explants, strongly express the

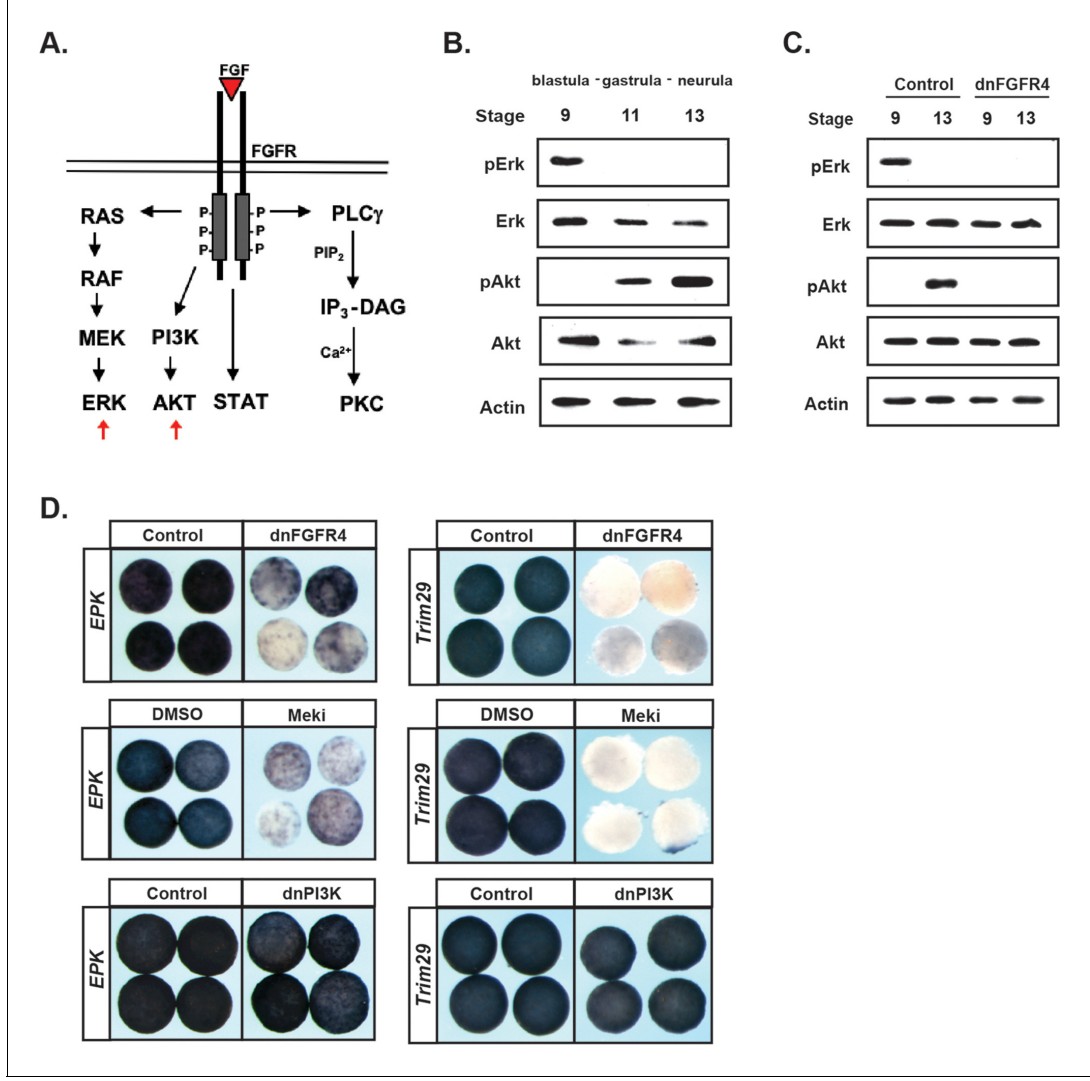

**Figure 3.** FGF signaling directs the transit from a pluripotent to a lineage restricted state through regulation of Erk and Akt activation. (**A**) Schematic representation of the FGF receptor and select signaling cascades activated downstream, highlighting the Ras/MAPK (Erk) and PI3K/Akt cascades. (**B**) Western blot of lysates from animal pole explants cultured alongside sibling embryos and collected at blastula (stage 9), midgastrula (stage 11), and early neurula (stage 13) stages to examine levels of phosphorylated and unphosphorylated Erk1/2 and Akt. Pluripotent cells show high pErk while lineage restricted cells display high pAkt. (**C**) Western blot of lysates from animal pole explants injected with dnFGFR4. Explants were cultured alongside sibling embryos and collected at blastula (stage 9) and early neurula (stage 13) stages to examine levels of phosphorylated and unphosphorylated Erk1/2 and Akt. Both pErk and pAkt are blocked by dnFGFR4. (**D**) Animal pole explant assay examining *Epidermal Keratin (EPK)* and *Trim29* expression in explants injected with either dnFGFR4 or dominant- negative PI3K (dnPI3K) or treated with Meki (RDEA119) and collected alongside sibling embryos at early neurula stages (stage 13–14). Meki treatment phenocopies dnFGFR4.

DOI: https://doi.org/10.7554/eLife.33845.005

The following figure supplement is available for figure 3:

**Figure supplement 1.** Meki (RDEA119) and PI3Ki (LY294) block activation of the MAPK and Akt cascades respectively.

DOI: https://doi.org/10.7554/eLife.33845.006

neuronal progenitor markers *Sox2* and *Sox3* at Stage 13, and the definitive neural marker *Nrp1* at stage 18. Inhibition of the PI3 Kinase/Akt cascade prevented cells from adopting a neuronal state in response to Chordin, as evidenced by decreased expression of *Sox2* (92%, n = 26), *Sox3* (100%, n = 28), and *Nrp1* (96%, n = 28) (*Figure 4b*). Similar results were obtained by blocking the PI3 Kinase/Akt cascade with the PI3 Kinase inhibitors LY294 and Wortmannin (*Figure 4—figure supplement 1b,c*). By contrast, blocking the Map Kinase cascade with RDEA119 did not interfere with expression of *Sox3* (93%, n = 27) or *Nrp1* (100%, n = 25), suggesting that this pathway is not

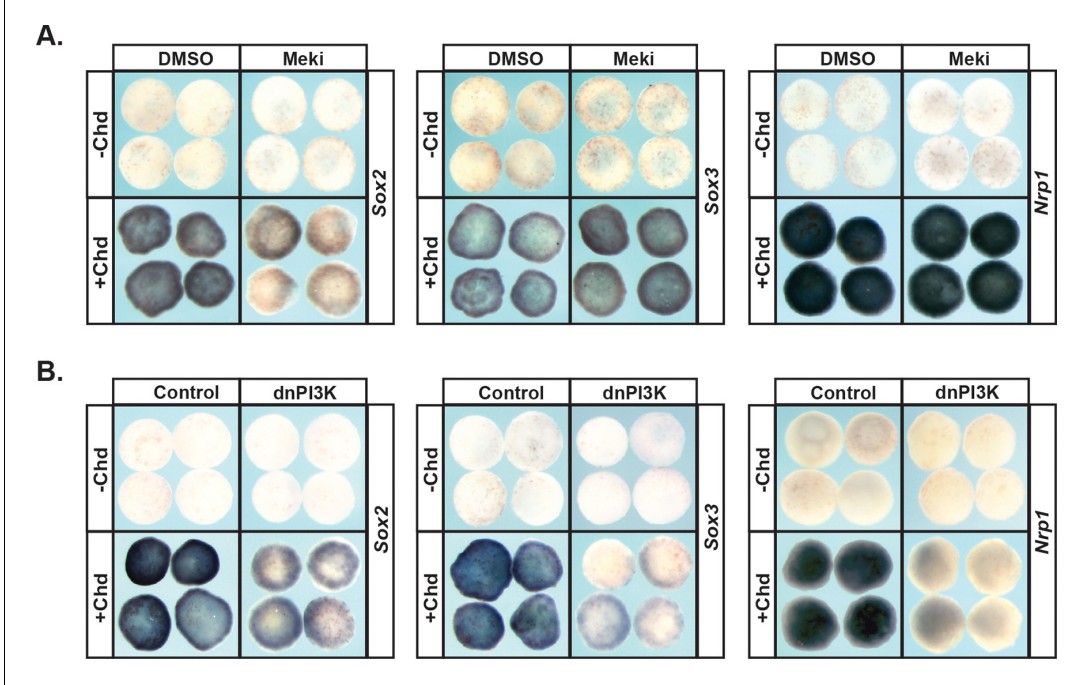

**Figure 4.** PI3K/Akt signaling but not MAPK signaling is required for pluripotent blastula cells to transit to a neural progenitor state. (A–B) Animal pole explant assay examining *Sox2*, *Sox3*, and *Nrp1* expression in Chordin (Chd) induced animal cap explants treated with Meki (RDEA119) (A) or injected with dnPI3K (B). Explants were cultured alongside sibling embryos and collected at early neurula stages (stage 13) for *Sox2/3* or late neurula stages (stage 18) for *Nrp1*. Meki treatment does not affect Chordin-mediated neural induction whereas dnPI3K blocks induction of all three neural markers.
DOI: https://doi.org/10.7554/eLife.33845.007

The following figure supplement is available for figure 4:

**Figure supplement 1.** Blocking PI3K/Akt activation using PI3Ki (L) (LY294) or PI3Ki (W) (Wortmannin) phenocopies the effects of dnPI3K on Chordin-mediated neural induction.
DOI: https://doi.org/10.7554/eLife.33845.008

essential for neural fates (*Figure 4a*). Similarly, blocking the Map Kinase cascade using a dominant-negative Raf1 (dnRaf) did not block Chordin-mediated neural induction (*Figure 4—figure supplement 1a*). Interestingly, we found that RDEA119 could interfere with Chordin-mediated *Sox2* expression in response to low levels of Chordin (92%, n = 26) but not high (*Figure 4a* and not shown).

## MAPK signaling and PI3K/Akt signaling are differentially required for transit to non-ectodermal lineages

Pluripotent animal pole cells can adopt mesodermal and endodermal fates, in addition to ectoderm-derived fates, under appropriate inducing conditions. Given our findings that Map Kinase and PI3 Kinase/Akt signaling are differentially required for these cells to transit to epidermal versus neuronal progenitor states, we further investigated the roles of these pathways in the formation of mesoderm and endoderm. The TGF-beta signaling pathway plays a central role in the formation of these two germ layers, and the ability of the ligand activin to induce mesodermal and endodermal states in a dose-dependent manner has been well documented (*Green and Smith, 1990*; *Thomsen et al., 1990*; *Hudson et al., 1997*). Treatment of control animal pole explants with low levels of activin is sufficient to promote a mesodermal state, as evidenced by high levels of *Xbra* and *MyoD*. We found that blocking activation of Map Kinase in these explants led to a complete loss of this mesodermal gene expression (100%, n = 60), as did blocking PI3 Kinase/Akt signaling (98%, n = 56), demonstrating that both of these signaling pathways play essential roles in the adoption of a mesodermal fate (*Figure 5a*), which is consistent with previous findings (*Umbhauer et al., 1995*; *Carballada et al., 2001*).

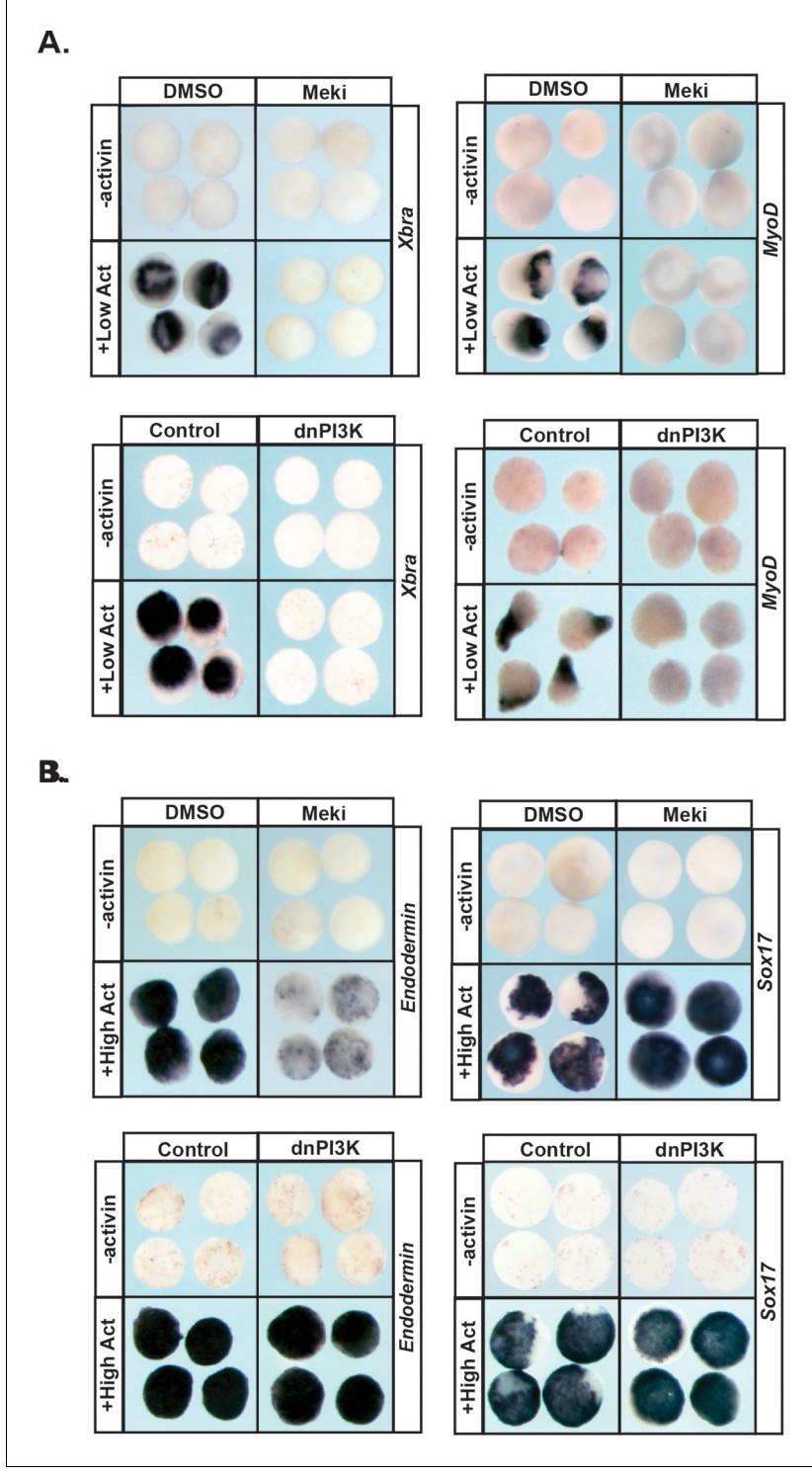

**Figure 5.** MAPK and PI3K/Akt differentially alter the transit of pluripotent cells to restricted cell states. (**A–B**) Animal pole explant assay examining expression of *Xbra* and *MyoD* (**A**) or *Endodermin* and *Sox17* (**B**) in explants cultured with or without activin after treatment with Meki (RDEA119) or injection with dnPI3K. Explants were cultured alongside sibling embryos and collected at midgastrula stages (stage 11.5) for *Xbra*, *Endodermin*, and *Sox17* expression and midneurula stages (stage 15/16) for *MyoD* expression. Blocking either cascade interferes with mesoderm formation whereas only MAPK signaling is required for *Endodermin* induction.
DOI: https://doi.org/10.7554/eLife.33845.009

The following figure supplements are available for figure 5:

*Figure 5 continued on next page*

*Figure 5 continued*

**Figure supplement 1.** MAPK and PI3K/Akt signaling are differentially required for *Endodermin* expression.
DOI: https://doi.org/10.7554/eLife.33845.010
**Figure supplement 2.** Blocking MAPK or PI3K activation differentially alters animal pole explant gene expression
(A–B) qRT-PCR analysis of animal cap explants treated with Meki (RDEA119) and cultured alongside sibling
embryos collected at blastula stages (stage 9) (A) or treated with PI3Ki (LY294) and cultured alongside sibling
embryos collected at early neurula stages (stage 13) (B).
DOI: https://doi.org/10.7554/eLife.33845.011

Treating pluripotent animal pole cells with higher doses of activin can induce endoderm formation, accompanied by expression of the primitive endodermal markers *Endodermin* and *Sox17*. We found that blocking PI3 Kinase/Akt activation had no effect on activin-mediated induction of *Endodermin* or *Sox17* in this assay, indicating that this signaling cascade is not essential for transit to an endodermal state (*Figure 5b*; *Figure 5—figure supplement 1a*). By contrast, inhibition of Map Kinase activation in these explants led to a loss of *Endodermin* expression (88%, n = 58), suggesting an inability to adopt a proper endodermal fate. Interestingly, expression of *Sox17* was increased following RDEA119 treatment (100%, n = 25), showing differential regulation of these two key markers of primitive endoderm (*Figure 5b*; *Figure 5—figure supplement 1a*).

## Prolonged MAPK activation alters the timing of pluripotency gene expression

A synthesis of the above findings indicates that blastula animal pole cells cannot adopt epidermal, mesodermal, or endodermal states when Map Kinase signaling is blocked, and are partially impaired in transiting to a neuronal progenitor state. We interpret these findings to mean that Map Kinase signaling is essential to the pluripotency of these cells. Consistent with this interpretation, qRT-PCR analysis of blastula stage explants treated with RDEA119 showed a significant reduction in expression of the pluripotency and blastula stage markers *Snail1*, *FoxD3*, *Zic1*, and *Sox2*, compared to control explants (*Figure 5—figure supplement 2a*). By contrast, while inhibition of PI3 Kinase/Akt signaling prevents transit to a neural or mesodermal state, it has no effect on the ability of pluripotent blastula cells to form endoderm or epidermis. Thus, this signaling pathway appears to be essential for transit to a subset of restricted states . To further understand the role of PI3 Kinase/Akt signaling in these cells, we examined the changes in gene expression elicited by treatment of explants aged to stage 13 in the presence of the PI3 Kinase inhibitor LY294. We found that when PI3 Kinase/Akt activation was blocked, a diverse set of lineage markers were up-regulated, including *Zic2*, *Sox3*, Sox17, and *MyoD*, potentially impeding adoption of certain lineage fates (*Figure 5—figure supplement 2b*). Importantly, the differential requirements for these two signaling cascades correlates with their temporal activation in animal pole cells, with high levels of pErk and low/absent pAkt characterizing pluripotent cells, whereas a transition to high pAkt and low pErk accompanies lineage restriction.

These findings suggest that prolonged Map Kinase signaling might interfere with the transition from a pluripotent to a lineage-restricted state. To examine this we used a constitutively active version of the upstream Mek kinase (Act-Mek) to activate Map Kinase (*Fukuda et al., 1997*) and examined the effects on expression of the pluripotency marker *Sox3* as cells transited from a pluripotent (St.9) to a lineage restricted (St.13) epidermal state. Prolonged activation of Map Kinase signaling was sufficient to maintain *Sox3* expression past the time when it would normally be down-regulated as cells lose pluripotency (92%, n = 49), leading to a 2–4 fold increase in *Sox3* expression over developmental time (*Figure 6a,b*). This suggests that prolonged Map Kinase signaling may delay the ability of these cells to exit the pluripotent state.

We also examined the effects of prolonged Map Kinase activity on the ability of blastula animal pole cells to adopt mesodermal or endodermal fates. We found that activation of the Map Kinase cascade alone also caused low-level expression of mesodermal markers (89%, n = 28), consistent with previous reports (*LaBonne et al., 1995*), and did not interfere with activin-mediated mesoderm formation. Interestingly, activating Map Kinase did interfere with transit to an endodermal state in response to high activin (81%, n = 27) (*Figure 6c*). By contrast, premature activation of PI3 Kinase/Akt activity (achieved by expressing a constitutively active p110 subunit of PI3 Kinase (p110caax)

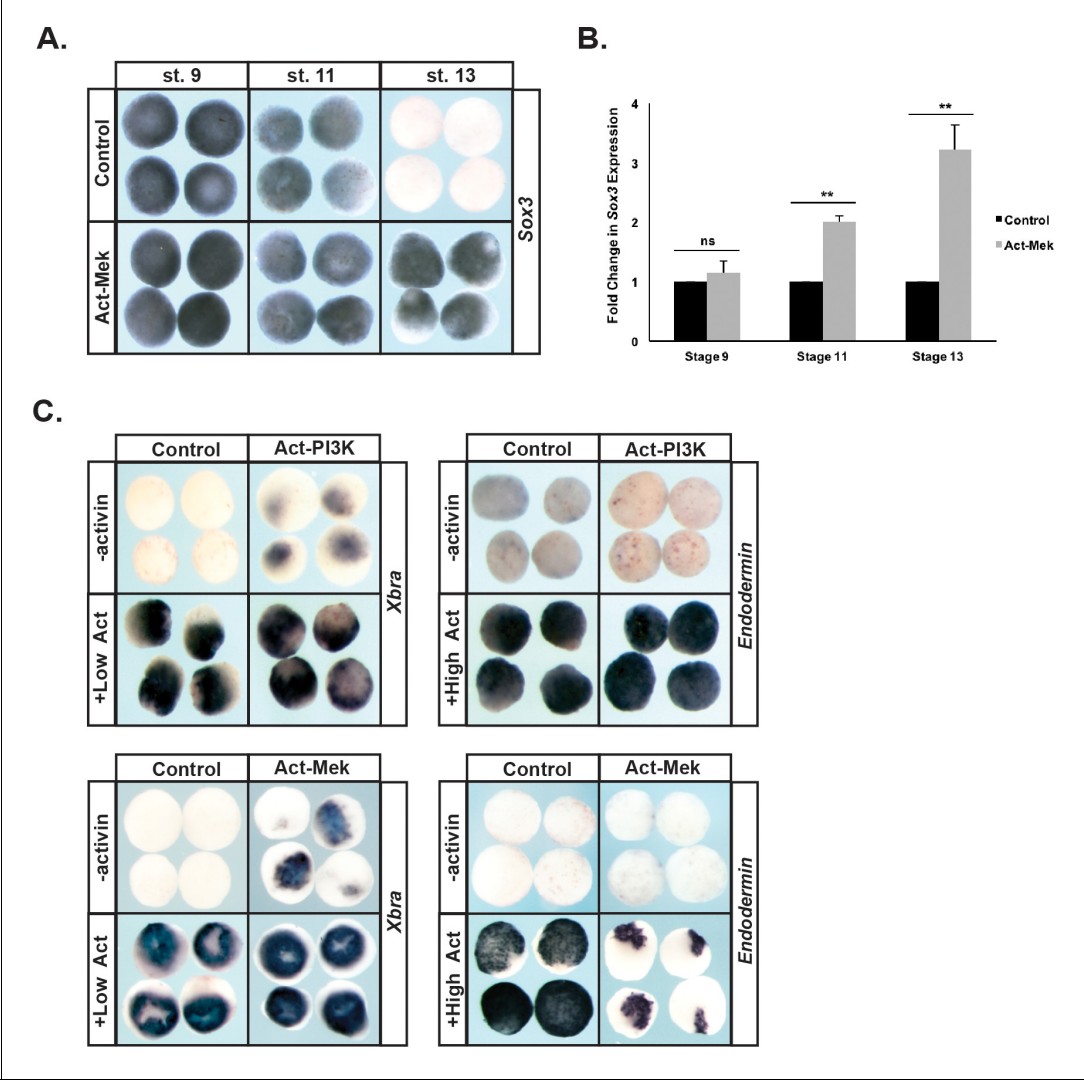

**Figure 6.** Prolonged MAPK activation alters the timing of pluripotency gene expression. (A–B) Animal pole explant assays examining *Sox3* expression in animal cap explants injected with constitutively active Mek (Act-Mek). Explants were cultured alongside sibling embryos and collected at blastula (stage 9), midgastrula (stage 11), and early neurula (stage 13) stages for in situ hybridization (A) or qRT-PCR (B). Activating MAPK leads to retained *Sox3* expression. (C) Animal cap explant assay examining *Xbra* and *Endodermin* expression in explants cultured with or without activin after injection with constitutively active PI3K (Act-PI3K) or Act-Mek. Explants were cultured alongside sibling embryos and collected at midgastrula stages (stage 11.5). Sustained MAPK activity interferes with *Endodermin* induction. (ns, not significant; **p<0.01).
DOI: https://doi.org/10.7554/eLife.33845.012

[*Carballada et al., 2001*; *Nie and Chang, 2007*]) did not affect the ability of blastula stem cells to transit to either a mesodermal or endodermal lineage in response to activin treatment, and indeed caused low-level activation of the mesodermal marker *Xbra* in the absence of activin (50%, n = 30) (*Figure 6c*).

## Reprograming blastula stem cells to a neural crest state leads to prolonged MAPK activation at the expense of PI3K activity

We recently proposed that Neural Crest cells arise via retention of the circuitry of pluripotency possessed by their blastula ancestors (*Buitrago-Delgado et al., 2015*; *Hoppler and Wheeler, 2015*). Intriguingly, our current work indicates that the pluripotent state is characterized by high Map Kinase activity, and low Akt signaling. This raises the important question of whether FGF-mediated activation of Map Kinase activity may contribute to establishment of the Neural Crest state in pluripotent

blastula cells, protecting them from lineage restriction, and similarly if activation of PI3 Kinase/Akt might oppose formation of the Neural Crest. To test this, we first asked if Map Kinase activity was required for establishment of the neural crest stem cell population at the neural plate border. When Map Kinase activation was blocked by expressing dnRaf, expression of neural crest markers *FoxD3*, *Sox9* and *Snail2* was lost. By contrast, blocking PI3 Kinase activation using the inhibitor Wortmannin did not significantly alter neural crest factor expression, despite completely blocking Akt activation (*Figure 7—figure supplement 2a,d*).

To further examine the link between establishment of the Neural Crest state and the balance between the Map Kinase and PI3 Kinase/Akt cascades, we asked if reprogramming cells to a Neural Crest state would alter the activity of these two signaling cascades. Animal pole explants can be reprogramed to a Neural Crest state by forced expression of the neural plate border factors Pax3 and Zic1 (*Monsoro-Burq et al., 2005*; *Hong and Saint-Jeannet, 2007*). Strikingly, high levels of pErk activity were maintained in these explants through stages when control explants are undergoing lineage restriction and adopting an epidermal state (*Figure 7a*). Similarly, reprogramed explants did not exhibit the increase in pAkt characteristic of the lineage-restricted state. These findings demonstrate that pluripotent blastula cells and Neural Crest cells share a common signature with respect to the activity of these two signaling cascades. Establishing a Neural Crest state was also accompanied by sustained expression of *FGFR4* (96%, n = 27) (*Figure 7—figure supplement 1a*).

The above findings suggest that FGF signaling, and the differential utilization of Map Kinase and PI3 Kinase/Akt activation in pluripotent vs. lineage restricted cells, could play a role in the retention of stem cell attributes underlying the establishment of the Neural Crest state. We hypothesized that

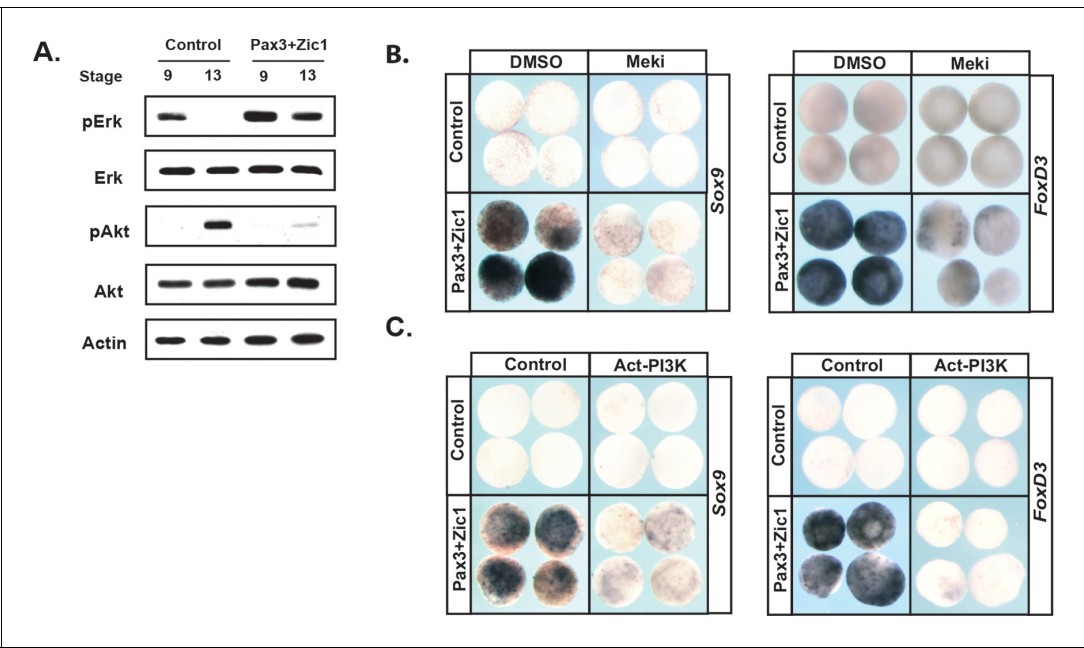

**Figure 7.** Reprograming cells to a Neural Crest state establishes and requires high MAPK and low PI3K/Akt activity. (**A**) Western blot of lysates from Pax3-GR/Zic1-GR injected animal pole explants. Explants were cultured alongside sibling embryos and collected at blastula (stage 9) and early neurula (stage 13) stages to examine levels of phosphorylated and unphosphorylated Erk1/2 and Akt. Reprograming to a neural crest state retains the activities of these pathways characteristic of pluripotent blastula cells. (**B–C**) Animal cap explant assay examining *Sox9* and *FoxD3* expression in Pax3GR/Zic1-GR injected explants treated with Meki (RDEA119) (**B**) or co-injected with Act-PI3K (**C**). Explants were cultured alongside sibling embryos and collected at late neurula stages (stage 18). Blocking MAPK activation or activating PI3K/Akt blocks expression of neural crest markers.
DOI: https://doi.org/10.7554/eLife.33845.013

The following figure supplements are available for figure 7:

**Figure supplement 1.** Reprograming to a Neural Crest state sustains *FGFR4* expression.
DOI: https://doi.org/10.7554/eLife.33845.014

**Figure supplement 2.** MAPK activation is required for neural crest factor expression at mid-neurula stages.
DOI: https://doi.org/10.7554/eLife.33845.015

if this were the case, then either blocking Map Kinase signaling or prematurely activating PI3 Kinase/Akt might block formation of the Neural Crest. To test this, we again used Pax3/Zic1-mediated reprogramming to establish the Neural Crest state in explants, which leads to robust expression of the Neural Crest markers in these cells at stage 18 (*Figure 7b,c*). Notably, blocking Map Kinase signaling in these explants with RDEA119 interfered with establishing a Neural Crest state, as evidenced by reduced expression of both *FoxD3* (88%, n = 26) and *Sox9* (85%, n = 27) (*Figure 7b,c*). Similar results were found following forced activation of PI3 Kinase/Akt using Act-PI3K (*FoxD3*: 81%, n = 26; *Sox9*: 81%, n = 27). These findings provide strong evidence that retention of blastula-stage potential in the cells that will ultimately become the Neural Crest is controlled, at least in part, by retaining the high Map Kinase:low PI3 Kinase/Akt signaling profile essential to the pluripotency of blastula animal pole cells (*Figure 8*).

## Discussion

Early embryonic cell fate decisions result from the interplay of a relatively small number of signaling pathways. Because these signals must be used reiteratively to direct a diverse array of outcomes, their output must be highly context-specific. Yet, in many cases, the mechanisms by which this is accomplished remain poorly understood. In this study, we uncover a striking switch in FGFR effector pathway utilization as cells transit from a pluripotent to a lineage-restricted state, adding important new insights into how FGF signaling regulates developmental potential. FGF-mediated Map Kinase activation is prominent in blastula stem cells prior to their exit from the pluripotent state, but then decreases as cells become lineage restricted. By contrast, FGF-mediated PI3 Kinase/Akt signaling is

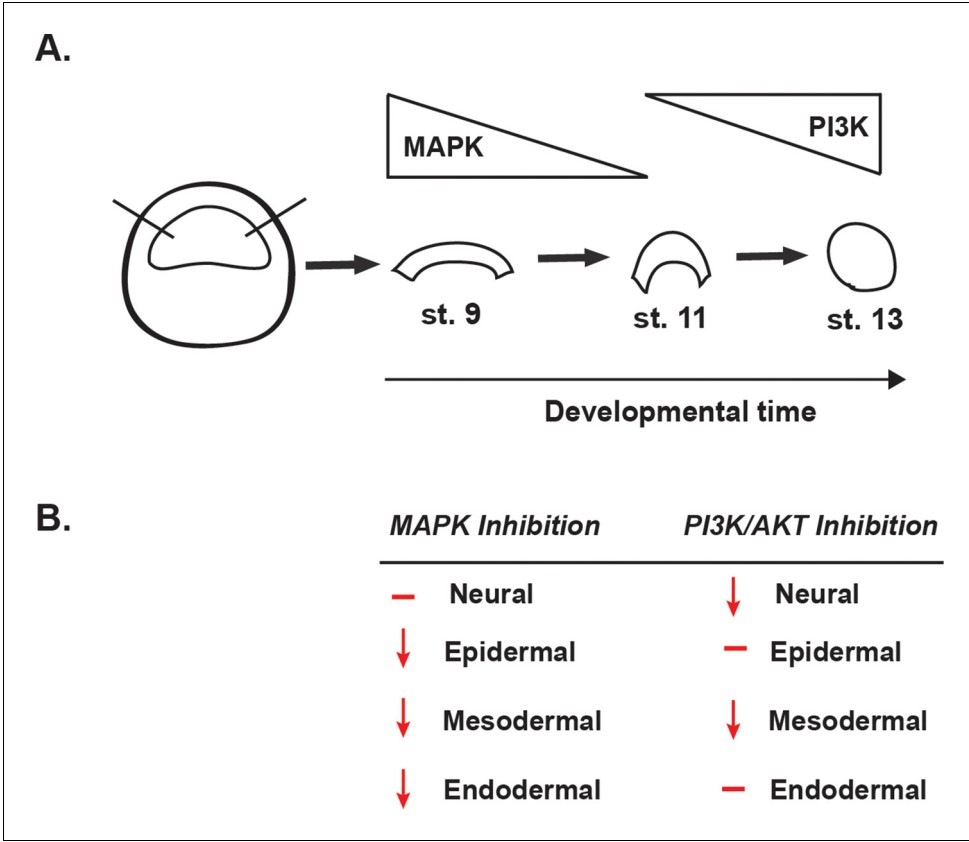

**Figure 8.** Summary of the effects of MAPK or PI3K/Akt inhibition. (**A**) Schematic representation of MAPK and PI3K/Akt cascade activation in animal cap explants (staged by sibling embryos) at the blastula stage (stage 9), midgastrula stage (stage 11), and early neurula stage (stage 13). (**B**) Diagram summarizing the effects of MAPK or PI3K/Akt inhibition on the adoption of neural, epidermal, mesodermal, and endodermal states.
DOI: https://doi.org/10.7554/eLife.33845.016

low in pluripotent cells, but increases dramatically as cells undergo lineage restriction. Importantly, both signaling cascades are blocked by dnFGFR4, showing that they are FGF mediated signals. However, although FGFR4 is the predominant FGF receptor expressed in blastula animal pole cells, it is not the only FGF receptor expressed there. It is possible that dnFGFR4 is also blocking the activity of other FGF receptors, such as FGFR1, as dominant negative proteins can sometimes inhibit the activity of related factors. We therefore interpret our findings using this receptor as demonstrating a role for FGF signaling in general, and not a specific role for FGFR4. Interesting, however, dnFGFR1 does not interfere with expression of neural crest markers in the manner that dnFGFR4 does (*Figure 1—figure supplement 1*).

Given the striking switch in cascade activation as cells move from pluripotency to lineage restriction, we chose to focus on the roles played by these signaling cascades. We show that Map Kinase signaling is essential for the pluripotency of blastula stem cells, whereas PI3 Kinase/Akt signaling appears to play a role in the ability of these cells to adopt a subset of lineage restricted states. Crosstalk between the Map Kinase and PI3 Kinase/Akt pathways is known to be highly context dependent, and can be cooperative or antagonistic (*Aksamitiene et al., 2012*). Indeed, antagonism between these pathways has been proposed to control the decision by angioblast progenitors to adopt artery versus venous fates (*Hong et al., 2006*). It will therefore be of great interest to investigate the mechanisms via which the observed switch in pathway utilization is achieved in blastula stem cells. Our preliminary analysis of the gene expression changes that occur as cells progress from a pluripotent to a lineage restricted state suggests that this might be mediated, at least in part, by a change in expression of intracellular adaptor proteins that scaffold the Map kinase vs. PI3 Kinase cascades (Geary and LaBonne, unpublished). We cannot rule out, however, that there could be a switch in the utilization of different FGF receptors that may contribute to the change in pathway utilization. Further studies may shed light on the very different effects that FGF signaling has on the pluripotency of naïve vs. primed embryonic stem cell cultures (*Brons et al., 2007*; *Kunath et al., 2007*; *Tesar et al., 2007*; *Hanna et al., 2010*; *Lanner and Rossant, 2010*).

We interpret the findings reported here as evidence that in this system, FGF-mediated Map Kinase activity is required for the pluripotency of blastula stem cells. An alternative interpretation might be that these signals are required for exit from pluripotency. We favor the first interpretation for a number of reasons including the observation that pMap Kinase is high when these cells are pluripotent and down-regulated as they exit pluripotency, and that activating MEK prolongs the expression of the pluripotency marker *Sox3*. Moreover, blocking Map Kinase signaling does not prevent transit to a neuronal progenitor state, which it might if these signals controlled exit from pluripotency.

In *Xenopus*, FGF signaling had previously been implicated in the adoption of both mesodermal and neural states, and our current work lends new mechanistic insights into these studies. For example, FGF signaling has been shown to be required for activin-mediated mesoderm formation, but its contributions to this process remained unclear (*LaBonne and Whitman, 1994*). Our findings that FGF-mediated Map Kinase activation is required for the pluripotency of blastula animal pole cells provides a long sought explanation for this requirement. Our work also sheds light on the role of FGF signaling in neural induction mediated by BMP-antagonists (*Launay et al., 1996*; *Sasai et al., 1996*). It has been shown that blocking FGF signaling with the tyrosine kinase inhibitor SU5402 does not prevent adoption of a definitive neural state in response to noggin, even though it can alter *Sox2/3* expression (*Wills et al., 2010*). We find similar effects when blocking Map Kinase signaling, with Chordin-mediated *Sox2* expression blocked, but not *Nrp1* (*Figure 4*). Conversely, we find that blocking the PI3 Kinase/Akt cascade prevents transit to both a neural progenitor and definitive neural state., The latter finding confirms what has been described previously and attributed to PI3 Kinase/Akt regulating GSK3 and Wnt signaling (*Peng et al., 2004*). Although we cannot rule out the possibility of cross-talk between these two signaling pathways, we report an earlier role for PI3 Kinase/Akt in mediating establishment of the neural progenitor state, prompting further exploration into a potential mechanism for its function during early ectodermal patterning. This could be linked to a recently described role for Akt in controlling progenitor cell progression (*Pegoraro et al., 2015*). The later role for PI3 Kinase/AKT signaling is consistent with the findings of Hongo and colleagues that a dominant negative FGFR4 inhibits commitment of animal cap cells to a neuronal state in response to neural-inducing cues (*Hongo et al., 1999*).

FGF signaling has also been previously implicated in the genesis of the neural crest (*LaBonne and Bronner-Fraser, 1998*; *Monsoro-Burq et al., 2003*; *Hong et al., 2008*; *Nichane et al., 2010*; *Garnett et al., 2012*), although its role relative to BMP and Wnt signaling has remained unclear. Neural Crest stem cells are of central importance to the development and evolution of vertebrates (*Groves and LaBonne, 2014*), and thus understanding the signals controlling their remarkable developmental potential is essential. Our recent work provides evidence that neural crest cells arise through partial retention of the regulatory network controlling the pluripotency of blastula ancestors (*Buitrago-Delgado et al., 2015*). Thus, it is crucial to re-examine the roles of signals, such as FGF, which had previously been hypothesized to 'induce' developmental potential in these cells, and ask if they might instead be acting earlier in development to control whether blastula animal pole cells become lineage restricted or retain pluripotency. Our findings that FGF-mediated Map Kinase signaling is required for the pluripotency of blastula animal pole cells supports such a model.

These findings led us to ask whether the striking change in FGF effector pathway utilization, from Map Kinase to PI3 Kinase/Akt, as cells transit from a pluripotent to a lineage restricted state could be important for the establishment of the Neural Crest population. Specifically, we wanted to know if the relative levels of Map Kinase and PI3 Kinase/Akt signals displayed by cells could predict or instruct the Neural Crest state. Strikingly, we found that cells that had been reprogrammed to a Neural Crest state by expression of Pax3 and Zic1 retained high levels of Map Kinase activity and low levels of Akt activity even at neurula stages, when control explants had become lineage restricted and transitioned to a high Akt, low Map Kinase activity state (*Figure 7a*). We found similar results when using an alternative regimen (Snail2 + Wnt signaling) for reprograming to a neural crest state (not shown) indicating that this signaling cascade signature correlates with the retention of pluripotency.

This correlation suggested that retaining a signature of high Map Kinase and low PI3 Kinase/Akt activity might be essential to establishing the Neural Crest stem cell population. We therefore asked if blocking Map Kinase activation and/or prematurely activating PI3 Kinase/Akt signals would interfere with reprograming animal pole explants to a Neural Crest state. Importantly, we found that the expression of *FoxD3* and *Sox9,* characteristic of the Neural Crest state established by Pax3/Zic1, was blocked by either inhibiting Map Kinase activity or prematurely activating Akt using a constitutively active PI3 Kinase subunit (*Figure 7b,c*). These data support a model in which context dependent control of effector pathways activated by FGF signaling in blastula animal pole cells controls not only the timing of the progression from pluripotency to lineage restriction of these cells, but also the retention of pluripotency and protection from lineage restriction in the cells that will become the Neural Crest. Our findings further suggest that the retention of FGF-mediated Map Kinase signaling in a subset of pluripotent blastula cells may have been an important step in the acquisition of Neural Crest cells, and thus in the evolution of vertebrates.

## Materials and methods

### Embryological methods

Wildtype *Xenopus laevis* embryos were collected at the indicated stages and processed for in situ hybridization as previously described (*LaBonne and Bronner-Fraser, 1998*). Manipulated whole embryos were microinjected into 1–2 cells at the 2–8 cell stage with mRNA (Ambion, mMessage mMachine SP6 Transcription Kit) as previously described (*Lee et al., 2012*) and collected at blastula stages (stage 9) or midneurula stages (stage 15–17) for in situ hybridization. Inhibitor-treated whole embryos were treated with the chemical inhibitor Wortmannin (Sigma) at a final concentration of 750 nM and collected at early neurula stages (stage 15) for in situ hybridization. Animal cap explants were manually dissected from wildtype or manipulated stage 9 embryos and aged to the denoted stage in 1xMMR. Manipulated embryos used for these animal cap dissections include embryos injected into both cells at the two-cell stage with the denoted mRNA and embryos treated at the 2–4 cell stage with a specific chemical inhibitor. For Map Kinase inhibition (Meki), the highly specific Mek1/2 chemical inhibitor Refametinib (RDEA119, Selleckchem) was used. Fresh RDEA119 (50–100 µM) was added to the culture media of explants upon dissection from RDEA119 treated embryos. For PI3 Kinase inhibition (PI3Ki), the chemical inhibitors LY294 (Sigma) and Wortmannin (Sigma) were

added to the culture media of explants dissected from stage 9 embryos. LY294 was used at a final concentration of 20 μM and Wortmannin was used at a final concentration of 100 nM. Both of these inhibitors can have off-target effects when used at higher doses. Pax3-GR and Zic1-GR explants were dissected from injected embryos treated with 15 μM Dexamethasone (Sigma) at stage 9 as previously described (*Buitrago-Delgado et al., 2015*). All results are representative of a minimum of three independent experiments.

## RNA isolation, CDNA synthesis, and qRT-PCR

RNA isolation, cDNA synthesis, and qPCR was performed as previously described (*Buitrago-Delgado et al., 2015*). Primers used include FoxD3, MyoD, ornithine decarboxylase (ODC), Sox2, Sox3, Sox11, Snail1, Sox17, Zic1, and Zic2 (sequences below). Expression was normalized to ODC and fold change calculated using ΔΔCT relative to stage 9 or stage 13 control samples. Represented is the mean of three independent biological replicates, with error bars depicting the standard error of the mean (SEM). An unpaired, two-tailed t-test was used to determine significance.

| Gene | Forward | Reverse |
| --- | --- | --- |
| FoxD3 | TCC TCT GAA CTG ACC AGG AA | TGC CGA CAC CCC AAT AAT GT |
| MyoD | CTG CTC CGA CGG CAT GAA | TCC CAA GTC TCA CGT CAT TG |
| ODC | TGA AAA CAT GGG TGC CTA CA | TGC CAG TGT GGT CTT GAC AT |
| Sox2 | TCA CCT CTT CTT CCC ATT CG | CGA CAT GTG CAG TCT GCT TT |
| Sox3 | CAC AAC TCG GAG ATC AGC AA | TCG TCG ATG AAG GGT CTT TT |
| Sox11 | GAA CTT CAC CCA GCA GAA CC | CCC TCG CTA CAA GAG TCC AA |
| Sox17 | GCA AGA TGC TTG GCA AGT CG | GCT GAA GTT CTC TAG ACA CA |
| Snail1 | AAG TCT CCC ATC AGC CCT TC | AGT CTT GCC CCC TTC ATC TT |
| Zic1 | CCT GGA TGT GGC AAA GTC TT | GTC ACA GCC TTC AAA CTC GC |
| Zic2 | AAT CCA CAA GAG GAC TCA CA | GTG TGC ACG TGC ATG TGC TT |

## Activin treatment of animal cap explants

Animal cap explants were isolated from control or manipulated blastula (stage 9) embryos. Following dissection, explants were cultured with recombinant Activin protein (R and D Systems) at a final concentration of 20–40 ng/mL for mesoderm induction and 100 ng/mL for endoderm induction in 1xMMR supplemented with 0.1% BSA as a carrier. Explants were cultured to midgastrula and midneurula stages (stage 11.5–16) following mesoderm induction and midgastrula stages (stage 11.5) following endoderm induction and processed for in situ hybridization.

## Western blot analysis

For western blot analyses, animal cap explants (10–20 explants) or whole embryos (five embryos) were lysed using a fresh 50 mM HEPES lysis buffer containing 5 mM EDTA, 2 mM Sodium Orthovanadate, 20 mM Sodium Fluoride, 10 mM β-Glycerophosphate, 1 mM Sodium Molybdate dihydrate, PhosStop phosphatase inhibitors (Roche), and protease inhibitors described previously (*Lee et al., 2012*). Animal cap explants were dissected from either wildtype or manipulated blastula (stage 9) embryos and cultured in 1XMMR until the indicated stage and collected. Stage 9 explants were collected 1 hr post-dissection. For the RDEA119 and LY294 time series, wildtype explants were dissected, cultured for 1 hr in 1X MMR for stage 9 treatment or cultured to stage 13, and subsequently cultured in inhibitor-containing media for the denoted length of time prior to collection. For Pax3-GR and Zic1-GR explant analysis, both control and Pax3-GR and Zic1-GR explants were cultured for 1 hr in 1XMMR, treated with 15 μM Dexamethasone, and then collected at the indicated stage. SDS-PAGE and Western blot analysis was used to visualize proteins, which were detected using the following antibodies: p44/42 MAPK (Erk1/2) (1:2000, Cell Signaling Technology), Phospho-p44/42 (Erk1/2) (Thr202/Tyr204) XP (1:2000, Cell Signaling Technology), Akt (1:2000, Cell Signaling Technology), Phospho-Akt (Ser473) XP (1:2000, Cell Signaling Technology), and Actin (1:8000, Sigma-

Aldrich,St. Louis, MO). Corresponding secondary antibodies conjugated to horseradish peroxidase (HRP) and chemiluminescense was used.

## DNA constructs

The truncated *Xenopus laevis* FGFR4 (AB007036) construct used (dnFGFR4) was cloned into a pCS2 vector from dnFGFR4-cs108, a kind gift from R. Harland (University of California, Berkeley). The dominant-negative PI3 Kinase subunit (dnPI3K, or Δp85) and constitutively-active PI3 Kinase subunit (Act-PI3K, or p110caax) was a gift from Chenbei Chang (University of Alabama), and constitutively-activate Mek (Act-Mek) was a gift from Ira Daar (National Cancer Institute, Maryland). Dominant-negative Raf (dnRaf) was generated by quick change mutagenesis (hRaf S621A) and subcloned into pCS2. All constructs received and cloned were confirmed by sequencing.

## Acknowledgements

We thank Joe Nguyen for invaluable technical assistance and members of the lab for helpful discussions. LG was supported by the NIH T32GM008061 and a Northwestern University Presidential Fellowship. This work was supported by NIH R01GM116538 to CL.

## Additional information

### Funding

| Funder | Grant reference number | Author |
| --- | --- | --- |
| National Institutes of Health | T32GM008061 | Lauren Geary |
| Northwestern University | Presidential Fellowship | Lauren Geary |
| National Institutes of Health | R01GM116538 | Carole LaBonne |

The funders had no role in study design, data collection and interpretation, or the decision to submit the work for publication.

### Author contributions

Lauren Geary, Data curation, Formal analysis, Investigation, Writing—original draft, Writing—review and editing; Carole LaBonne, Conceptualization, Formal analysis, Supervision, Funding acquisition, Writing—review and editing

### Author ORCIDs

Carole LaBonne [iD] http://orcid.org/0000-0001-6001-7179

### Ethics

Animal experimentation: This study was performed in strict accordance with the recommendations in the Guide for the Care and Use of Laboratory Animals of the National Institutes of Health. All of the animals were handled according to a protocol approved by the institutional animal care and use committee (IACUC) protocols at Northwestern University (Protocol # IS00001963 ).

### Decision letter and Author response

Decision letter https://doi.org/10.7554/eLife.33845.019
Author response https://doi.org/10.7554/eLife.33845.020

## Additional files

### Supplementary files

• Transparent reporting form
DOI: https://doi.org/10.7554/eLife.33845.017

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
