## [Decision Letter]

Editors’ note: a previous version of this study was rejected after peer review, but the authors submitted for reconsideration. The first decision letter after peer review is shown below.]

Thank you for choosing to send your work, "FGF mediated MAPK and PI3K/Akt Signals make distinct contributions to pluripotency and the establishment of Neural Crest", for consideration at *eLife*.

Your submission has been assessed by a Senior and Reviewing Editor, Marianne Bronner, and three reviewers who have elected to remain anonymous. The reviewers found the paper to be potentially interesting and well-written. However, they also felt the authors need to strengthen biochemical characterization of the pathway and confirm that this proposed "molecular switch" is downstream of the receptor. To accomplish this the authors would have to collect persuasive evidence linking Akt to FGF by demonstrating dnFGFR4 specificity, using other means of interfering with pathways downstream of FGF, and using more markers to prove their point. In addition, they should better discuss previous literature on this topic as well as alternative interpretations of their data. In general, it was felt that the present manuscript is a description of a phenomenon rather than a molecular characterization of a new mechanism. Please see the individual reviews below for further details.

Given that these revisions are likely to take a significant amount of time and the policy of *eLife* is to reject papers that require more than two months for modifications, we have no choice but to decline the present version of the paper. That said, if you feel you can make the revisions, we will make every effort to return a new version of the manuscript to the same reviewers.

*Reviewer #1:*

Geary and LaBonne describe differential activation of the downstream effector pathways of FGF-signaling in pluripotent and lineage-restricted *Xenopus* cells. They utilize animal cap assays to show that blastula stem cells have high levels of MAP Kinase signaling, while cells that are differentiating have prevalence of PI3 Kinase/Akt signals – both downstream of FGFs. The authors suggest that the transition between these two effector pathways, which takes place during gastrulation, is required for the establishment of distinct cell lineages. Furthermore, reprogramming experiments are employed to show that maintenance of high MAP Kinase signaling is required for the establishment of the neural crest population. In line with their previous work, the authors suggest that maintenance of FGF/MAPK signaling from blastula stages is necessary for specification of the neural crest and might underlie its multipotency.

The manuscript is well written and the results are presented clearly. The switch in FGF effector deployment is an exciting phenomenon that holds important implications for the mode of action of signaling pathways during early development. Whereas the authors convincingly tie this molecular switch to cellular properties such as multipotency, the fact that virtually all experiments are performed in animal cap assays raises substantial concerns about the validity of the authors' findings in vivo. Thus, evidence of the presence of the MAPK-PI3K switch in developing embryos would be an important addition to the work.

Major points:

1) Lack of in vivo experiments: The authors employ animal cap assays for both loss and gain of function experiments, and to test questions pertaining to cell potential of blastula and neural crest cells. Validation of the FGF effector switch in vivo would greatly enhance the work. In particular, I was puzzled by the reprogramming experiments – why not examine neural crest formation as it takes place in a developing embryo instead of artificially generating neural crest-like cells through overexpression?

2) On the central role ascribed to FGFR4: Throughout the text, the authors state that activation of the MAP Kinase and the PI3 Kinase/Akt pathways are downstream of FGFR4. This claim is based on the expression levels of FGF receptors in the early *Xenopus* embryo (data not shown), and the use of a FGFR4 dominant negative construct, which causes loss of both Erk and Akt phosphorylation. Given the promiscuity of dnFGFR constructs and the presence of multiple FGF receptors in the blastula/neural crest, I am not convinced that FGFR4 is the main trigger of the MAPK and PI3 effector pathways. Instead, I am tempted to speculate that the switch occurs due to activation of a different FGFR type or isoform. The central role of FGFR4 in this process can only be confirmed by specific knockdown of this transcript.

3) Crosstalk between FGF effector pathways: Since intracellular effector pathways of signaling systems are very complex and known to intersect, how does activation/inhibition of PI3K affect activity of MAPK and vice versa? Are the effector pathways inhibiting each other as one might speculate from Figure 3 and Table 1?

4) Possible mechanisms of the molecular switch: While characterization of the molecular mechanism controlling the switch between effector pathways may be beyond the scope of the work, as I reader I expected to find this point addressed in the Discussion. What would be the factors that could tilt the balance between MAPK and PI3K, specifically during gastrula stages?

*Reviewer #2:*

This manuscript makes advances in the understanding of pluripotency of the "animal cap" which here is referred to as pluripotent ectoderm, in line with the hypothesis that Neural crest is the retention of a pluripotent state. This case was made in a previous paper, and the current work builds on that.

This manuscript makes the case that FGF acting through FGFR4, and MAPkinase maintains pluripotency. In contrast, Akt/PI3Kinase terminate pluripotency.

The manuscript builds a circumstantial case for the importance of FGFR4 in maintaining pluripotency, but the results are based on a dominant negative receptor, whose specificity has not been proven. Therefore discussion regarding FGFR4 dominant negatives, and whether this manipulation implicated FGFR4 specifically in signaling needs to be tightened up or experimentally verified.

The results with FGFR4 appear to be very different from those from Hongo/Okamoto, who essentially argued the opposite, that DN FGFR4 blocks anterior neural induction. The precise experiments seem different, but it would be very useful to arrive at a coherent explanation if possible.

The most interesting aspect of the work is the distinction between Akt/PI3K activity in not being required for pluripotency, and Mapkinase activity requirement for pluripotency. One would expect that this would be more thoroughly documented, using independent inhibitors of the pathways, and assaying additional consequences of pluripotency, in addition to Epidermal Keratin. I don't think this would be an unusual request, but is in line with the general view that pharmacological manipulations, where possible should be backed up with additional drug or molecular manipulation.

With respect to the inhibition of progression of the cells to neural states with the PI3K inhibitor, it would be essential to know what the cells are doing instead (presumably retaining pluripotency markers?). It is important to rule out the possibility that this does not simply cause cell death, or general ill health of the cells, which would also prevent maturation of the cells, which some have argued accounts for the failure of neutralization when FGF is inhibited (Wills, Harland).

With respect to the requirement for both MapKinase and PI3Kinase pathways being required for mesoderm induction, the manuscript should refer to previous work on this, such as Umbhauer and Smith, and Carballada, Lemaire. Their results appear to be similar, so perhaps this is not new.

Figure 6 is not very compelling – the premature downregulation of Sox3 by p110-Caax and the retention of Sox3 by MAPK should be quantified more convincingly.

Figure 7 are not very compelling; again some quantification would be useful.

*Reviewer #3:*

In this manuscript, the authors investigate the signalling requirements for *Xenopus* animal caps to remain in a multipotent state or to differentiate into different ectodermal derivatives (epidermis/neural) or other germ layers (endoderm/mesoderm). FGF signalling is well known to be important for the regulation of pluripotency in mouse and other species, hence they focus on FGF.

The authors show that FGF signalling via FGFR4 is required for early blastula animal caps to become epidermis, and that as animal caps age they switch from high MAPK to high Akt signalling, while losing MAPK activity. While the acquisition of epidermal fate depends on MAPK signalling, the acquisition of neural identity after Chordin injection requires Akt, thus, both pathways are required differentially. They then go on to show that the same is true for the acquisition of endodermal (MAPK dependent) and mesoderm (Akt dependent) fates.

Next, they aim to assess whether prolonged MAPK or premature Akt activation alters animal cap 'pluripotency' (only assessed by Sox3) and suggest that this indeed changes the retention of pluripotent character.

Finally, the authors have previously shown that neural crest cells have retained pluripotency rather than undergoing lineage restriction. They now assess whether induction of neural crest cells by a combination of transcription factors alters MAPK or Akt signalling and show that this indeed activates MAPK, but not Akt, and that MAPK inhibition or Akt overactivation prevents neural crest induction by these factors.

They conclude that MAPK is essential for the pluripotency state, while Akt is important for animal caps to adopt some lineages.

I am somewhat concerned about the novelty of their findings and about the final interpretation of the results with a central conclusion relying on a single marker. Modulation of FGF signalling during pluripotency and the loss thereof is well established, and in this sense the study adds detail to a well-established principle rather than revealing novel concepts. Of course it is interesting how different downstream pathways act differentially, however this in itself is not a new principle.

A central part of their conclusion is that MAPK mediated FGF signalling is required for cells to maintain pluripotency (e.g. Discussion, second paragraph). However, they never show that pluripotency makers depend on MAPK – they only ever examine Oct60 and Sox3 with Sox3 also being expressed in neural tissue. This is not sufficient to make a general conclusion about pluripotency. In Figure 6 they show that prolonged MAPK activation leads to prolonged Sox3 expression, while activation of Akt does the opposite. They equate this to retaining pluripotency. However, as they themselves show in Figure 2, they consider Sox3 as neural marker and argue that this is indeed neural tissue because it does not express Oct60. They must demonstrate that in the experiment in Figure 6 Oct60 and other 'pluripotency makers' are affected and that MAPK inhibition leads to loss of pluripotency genes. For this to be convincing they have to demonstrate a panel of genes rather than just one.

The authors argue that retaining MAPK activity is key for retaining pluripotency in neural crest cells. However they do not show this; they show that neural crest induction by Pax3/Zic1 elevates FGFR4 expression and MAPK suggesting that these changes in signalling activity is a consequence of neural crest cell formation, but not the driver.

In Figure 2 the authors show that dnFGFR4 enhances neural markers (sox3, nrp1). They then go on to show that Akt signalling is required for cells to become neutralised when injected with chordin (Figure 4). These results are contradictory, and the first result is inconsistent with findings that FGF is required for neural induction. Does Akt or MAPK inhibition enhance neural gene expression similar to what is shown in Figure 2?

In summary, while the experiments are generally well conducted and controlled the results appear to be overinterpreted often depending on a single marker and the conclusions are rather forceful. The involvement of FGF in pluripotency is well established, although a potential differential role if this can be proven may be interesting.

[Editors’ note: what now follows is the decision letter after the authors submitted for further consideration.]

Thank you for submitting your article "FGF mediated MAPK and PI3K/Akt Signals make distinct contributions to pluripotency and the establishment of Neural Crest" for consideration by *eLife*. Your article has been reviewed by three peer reviewers, and the evaluation has been overseen by Marianne Bronner as the Senior and Reviewing Editor. The reviewers have opted to remain anonymous.

The reviewers have discussed the reviews with one another and the Reviewing Editor has drafted this decision to help you prepare a revised submission.

Summary:

The reviewers find the manuscript improved. However, they think that some of the conclusions overreach the data and that parts of the manuscript should be rewritten and softened, as described below:

Essential revisions:

1) Since the additional experiments examining the role of FGFR4 were not definitive, we ask that you rewrite this part of the manuscript to make it clear that a possible role for FGFR4 is speculative. While it may have a role in the process, the evidence is not definitive and the mechanism by which the DNFGFR1 construct has effects different from DNFGFR1 is unresolved.

2) It's unclear if the experiment with the DNFGFR1 is of any significance and feels like an afterthought. This would also impact the Discussion since it is possible that the switch between MAPK and PI3K involves activation of distinct receptors.

---

## [Author Response]

[Editors’ note: the author responses to the first round of peer review follow.]

Reviewer #1:[…] 1) Lack of in vivo experiments: The authors employ animal cap assays for both loss and gain of function experiments, and to test questions pertaining to cell potential of blastula and neural crest cells. Validation of the FGF effector switch in vivo would greatly enhance the work. In particular, I was puzzled by the reprogramming experiments – why not examine neural crest formation as it takes place in a developing embryo instead of artificially generating neural crest-like cells through overexpression?

These experiments were done in explants because in those you can monitor the progression from pluripotency to lineage restricted without influence from other tissues and signals, allowing the experiments to be more tightly controlled. This also allows separation from the confounded role of FGF signaling in posteriorization of the embryonic axes. In this work we were focused on how pluripotent blastula (ES) cells retain their pluripotency, and this was best explored by isolating them. Also, because pharmacological inhibitors lack spatial control, you cannot be certain that the phenotypic effects are direct on the tissue of interest when working with intact embryos. Nonetheless, in the revised manuscript we have now included whole embryo experiments as supplements to Figure 1 and Figure 7 confirming the effects of these pathways on neural crest endogenously.

2) On the central role ascribed to FGFR4: Throughout the text, the authors state that activation of the MAP Kinase and the PI3 Kinase/Akt pathways are downstream of FGFR4. This claim is based on the expression levels of FGF receptors in the early Xenopus embryo (data not shown), and the use of a FGFR4 dominant negative construct, which causes loss of both Erk and Akt phosphorylation. Given the promiscuity of dnFGFR constructs and the presence of multiple FGF receptors in the blastula/neural crest, I am not convinced that FGFR4 is the main trigger of the MAPK and PI3 effector pathways. Instead, I am tempted to speculate that the switch occurs due to activation of a different FGFR type or isoform. The central role of FGFR4 in this process can only be confirmed by specific knockdown of this transcript.

We agree that we have not ruled out the possibility that the dnFGFR4 does not block the activity of other FGFRs and we now discuss this in the text, and make clear that we are blocking FGF signaling, but not necessarily only FGFR4 signaling. We also include data showing that dnFGFR1 does not have the same effects on neural crest that dnFGFR4 does. We note that in response to the reviews we did attempt to morpholino deplete FGFR4 vs FGFR1 – we purchased four published MOs for these receptors listed on the Xenbase site, but we found they tended to crash out of solution and were not useful for loss of function assays. We asked Gene Tools to look at the sequences, and opined that they were poorly designed and predicted to be insoluble. Rather than spend more money on morpholinos, we have more recently been trying to use G0-CRiSPR mutagenesis to deplete these receptors. Unfortunately we have not been able to get early and uniform enough mutagenesis to do these experiments (plus the receptors are maternally provided) and we have delayed publication of this work as long as we can. We had never meant to emphasize the role of one particular receptor in this work (we used FGFR4 because this receptor is the predominant one expressed at blastula stages) – our focus has always been on the downstream effector pathways of FGF signaling and their functions and we make that clear in the text.

3) Crosstalk between FGF effector pathways: Since intracellular effector pathways of signaling systems are very complex and known to intersect, how does activation/inhibition of PI3K affect activity of MAPK and vice versa? Are the effector pathways inhibiting each other as one might speculate from Figure 3 and Table 1?

Figure 3—figure supplement 1 now shows the specificity of the Mek and PI3K inhibitors for their own pathways.

4) Possible mechanisms of the molecular switch: While characterization of the molecular mechanism controlling the switch between effector pathways may be beyond the scope of the work, as I reader I expected to find this point addressed in the Discussion. What would be the factors that could tilt the balance between MAPK and PI3K, specifically during gastrula stages?

We now address this point in the Discussion. We believe that the context/competency of the cells changes over time, and that changes in the expression of intracellular FGF pathway adaptors/modulators over developmental time likely underlies the switch in effector pathway utilization. We agree that experiments on this are beyond the scope of the current work.

Reviewer #2:[…] The manuscript builds a circumstantial case for the importance of FGFR4 in maintaining pluripotency, but the results are based on a dominant negative receptor, whose specificity has not been proven. Therefore discussion regarding FGFR4 dominant negatives, and whether this manipulation implicated FGFR4 specifically in signaling needs to be tightened up or experimentally verified.The results with FGFR4 appear to be very different from those from Hongo/Okamoto, who essentially argued the opposite, that DN FGFR4 blocks anterior neural induction. The precise experiments seem different, but it would be very useful to arrive at a coherent explanation if possible.

The reviewer refers to a 1999 DB paper that focused on a role for FGF signaling in anterior neural induction that used dnFGFR1 and dnFGFR 4 constructs. The reviewer is correct that the experiments in that paper are done quite differently than ours: explants were isolated at gastrula stages (stage 10), a number of the experiments were done with dissociated cells, and changes in gene expression were largely analyzed by RT-PCR assays at stage 25 with only two genes in common to our study. With so many differences it’s how to come up with a single explanation for why the results differ. We can only report the results of our experiments, which focused on earlier stages and which were all carried out numerous times with highly reproducible results.

The most interesting aspect of the work is the distinction between Akt/PI3K activity in not being required for pluripotency, and Mapkinase activity requirement for pluripotency. One would expect that this would be more thoroughly documented, using independent inhibitors of the pathways, and assaying additional consequences of pluripotency, in addition to Epidermal Keratin. I don't think this would be an unusual request, but is in line with the general view that pharmacological manipulations, where possible should be backed up with additional drug or molecular manipulation.

In the revised manuscript we have repeated all of the experiments with at least one additional way of inhibiting each pathway. We now use dnRaf, in addition to the Mek inhibitors, for Mapk inhibition. We use Wortmanin and Ly294, in addition to dnPI3K, for Akt inhibition. We also use multiple markers for each cell fate examined (EpK and Trim29 for epidermis; Xbra and MyoD for mesoderm; Sox17 and endodermin for endoderm; *Sox2*/3 and Nrp1 for neural etc.). We examine pluripotency both by marker expression and by functional tests of pluripotency. We also now include qPCR data quantifying gene expression changes in response to inhibition of each pathway.

With respect to the inhibition of progression of the cells to neural states with the PI3K inhibitor, it would be essential to know what the cells are doing instead (presumably retaining pluripotency markers?). It is important to rule out the possibility that this does not simply cause cell death, or general ill health of the cells, which would also prevent maturation of the cells, which some have argued accounts for the failure of neutralization when FGF is inhibited (Wills, Harland).

The revised manuscript includes qPCR data examining the changes in gene expression in explants inhibited for each pathway. In the case of the PI3K inhibitor there is an interesting up-regulation of a broad set of lineage markers, suggesting that the cells are poised for lineage decisions but impaired in their decision making. We were also very careful to use low doses of inhibitors such as LY294 as they can have off target effects at higher doses. For our studies we titrated our doses to the lowest level sufficient to completely block Akt activation, which is 3-10 fold lower than doses used in some studies and which can generate nonspecific effects, so we are confident that these changes in gene expression are specific to this pathway.

With respect to the requirement for both MapKinase and PI3Kinase pathways being required for mesoderm induction, the manuscript should refer to previous work on this, such as Umbhauer and Smith, and Carballada, Lemaire. Their results appear to be similar, so perhaps this is not new.

These papers are now cited/discussed. These pathways have indeed been implicated in mesoderm induction previously. What differs here is how that finding is interpreted. Those studies, and those of LaBonne and Whitman, focused on a single lineage decision. Our current work is focused on pluripotency and lineage decisions more broadly. We thus provide a new interpretation of those prior findings and put them in a broader context. For example, with respect to Mapk, we propose a general effect on pluripotency rather than a specific effect on mesoderm.

Figure 6 is not very compelling – the premature downregulation of Sox3 by p110-Caax and the retention of Sox3 by MAPK should be quantified more convincingly.

We have both replaced the images and included qPCR data here.

Figure 7 are not very compelling; again some quantification would be useful.

We have provided better images for these data. Other typos and references errors noted in the review have been corrected and we thank the reviewer for prompting these changes, which have improved the manuscript.

Reviewer #3:[…] I am somewhat concerned about the novelty of their findings and about the final interpretation of the results with a central conclusion relying on a single marker. Modulation of FGF signalling during pluripotency and the loss thereof is well established, and in this sense the study adds detail to a well-established principle rather than revealing novel concepts. Of course it is interesting how different downstream pathways act differentially, however this in itself is not a new principle.

We respectfully disagree about the novelty of these findings. First, a role for FGF in pluripotencyhad not been shown in this system, which is the system in which a lot of prior work on germ layer formation had been carried out. Indeed FGF had been ascribed very different roles in the *Xenopus* system, which did not integrate very well with findings in mouse and human ES cells. We feel therefore that our results advance the field by showing the central role this signaling is playing in pluripotency is evolutionarily conserved. Our work also further advances the field by adding exciting and very novel data on a switch in effector pathways. This finding is not only important in its own right, but also may help explain differences in what FGF signaling does in naive vs. primed ES (and human vs. mouse) cells. Finally, we show a completely novel role for FGF dependent Map kinase signaling in the retention of pluripotency that led to neural crest formation, providing a significant mechanistic advance over our prior Science paper.

A central part of their conclusion is that MAPK mediated FGF signalling is required for cells to maintain pluripotency (e.g. Discussion, second paragraph). However, they never show that pluripotency makers depend on MAPK – they only ever examine Oct60 and Sox3 with Sox3 also being expressed in neural tissue. This is not sufficient to make a general conclusion about pluripotency. In Figure 6 they show that prolonged MAPK activation leads to prolonged Sox3 expression, while activation of Akt does the opposite. They equate this to retaining pluripotency. However, as they themselves show in Figure 2, they consider Sox3 as neural marker and argue that this is indeed neural tissue because it does not express Oct60. They must demonstrate that in the experiment in Figure 6 Oct60 and other 'pluripotency makers' are affected and that MAPK inhibition leads to loss of pluripotency genes. For this to be convincing they have to demonstrate a panel of genes rather than just one.

The strongest and most relevant data in assessing pluripotency comes from functional studies that challenge cells to adopt different fates/states. Those pluripotency experiments are shown in Figure 3, Figure 4 and Figure 5, and each fate/state is evaluated with more than one marker. We believe these data collective provide compelling evidence for a loss of pluripotency. In addition we examine changes in the expression of a panel of genes in response to inhibitor treatment in Figure 5—figure supplement 2.

The authors argue that retaining MAPK activity is key for retaining pluripotency in neural crest cells. However they do not show this; they show that neural crest induction by Pax3/Zic1 elevates FGFR4 expression and MAPK suggesting that these changes in signalling activity is a consequence of neural crest cell formation, but not the driver.

We do show that establishing a neural crest state using Pax3/zip retains pMapK (and FGFR4 expression), however we also more directly show in Figure 7 that blocking Map kinase activation blocks expression of neural crest markers *Sox9* and *Foxd3* in explants. In addition, in Figure 7—figure supplement 2, we show that dnRaf blocks neural crest formation in whole embryos.

In Figure 2 the authors show that dnFGFR4 enhances neural markers (sox3, nrp1). They then go on to show that Akt signalling is required for cells to become neutralised when injected with chordin (Figure 4). These results are contradictory, and the first result is inconsistent with findings that FGF is required for neural induction. Does Akt or MAPK inhibition enhance neural gene expression similar to what is shown in Figure 2?

The level of Sox3/Nrp1 expression found in dnFGFR4 explants is much weaker than observed in chordin-neuralized explants (now shown in Figure 2). We think this represents a bias toward a neuronal progenitor state but not neural induction. Blocking Mapk activation does not affect chordin-mediated neural induction, but blocking Akt activation does, as the reviewer notes. Interestingly blocking Akt also leads to weak upregulation of *Sox2* and Sox3 along with other lineage markers (see Figure 4—figure supplement 2), which we interpret as evidence that the cells are poised for lineage decisions, but impaired in their decision making. We don’t find these data contradictory but rather that it reflects the complexity of signaling systems, and blocking any one downstream signaling cascade may not fully recapitulate blocking all downstream signals. It should be noted that FGF signaling is not required for neural induction/anterior neural development in *Xenopus* (BMP antagonism is sufficient for this, see Wills et al. 2010 for example) but has been shown to be essential for posterior neural development.

[Editors' note: the author responses to the re-review follow.]

Essential revisions:1) Since the additional experiments examining the role of FGFR4 were not definitive, we ask that you rewrite this part of the manuscript to make it clear that a possible role for FGFR4 is speculative. While it may have a role in the process, the evidence is not definitive and the mechanism by which the DNFGFR1 construct has effects different from DNFGFR1 is unresolved.

We have rewritten these parts of the manuscript to reflect this.

2) It's unclear if the experiment with the DNFGFR1 is of any significance and feels like an afterthought. This would also impact the Discussion since it is possible that the switch between MAPK and PI3K involves activation of distinct receptors.

We have added this to the Discussion.